# Integrating Novel and Classical Prognostic Factors in Locally Advanced Cervical Cancer: A Machine Learning-Based Predictive Model (ESTHER Study)

**DOI:** 10.3390/jpm15040153

**Published:** 2025-04-15

**Authors:** Federica Medici, Martina Ferioli, Arina Alexandra Zamfir, Milly Buwenge, Gabriella Macchia, Francesco Deodato, Paolo Castellucci, Luca Tagliaferri, Anna Myriam Perrone, Pierandrea De Iaco, Lidia Strigari, Alberto Bazzocchi, Stefania M. R. Rizzo, Costanza Maria Donati, Alessandra Arcelli, Stefano Fanti, Alessio Giuseppe Morganti, Savino Cilla

**Affiliations:** 1Département de Radiothérapie, Gustave Roussy, 94 805 Villejuif, France; federica.medici@ausl.re.it; 2Radiation Oncology, IRCCS Azienda Ospedaliero—Universitaria di Bologna, 40138 Bologna, Italy; costanzamaria.donati@unibo.it (C.M.D.); alessandra.arcelli@aosp.bo.it (A.A.); alessio.morganti2@unibo.it (A.G.M.); 3Radiation Oncology, Azienda Ospedaliero—Universitaria di Ferrara, 44124 Ferrara, Italy; martina.ferioli@ospefe.it; 4Department of Medical and Surgical Sciences (DIMEC), Alma Mater Studiorum University of Bologna, 40138 Bologna, Italy; milly.buwenge2@unibo.it (M.B.); myriam.perrone@unibo.it (A.M.P.); pierandrea.deiaco@aosp.bo.it (P.D.I.); 5Radiotherapy Unit, Responsible Research Hospital, 86100 Campobasso, Italy; gabriella.macchia@responsible.hospital (G.M.); francesco.deodato@unicatt.it (F.D.); 6Nuclear Medicine, IRCCS Azienda Ospedaliero—Universitaria di Bologna, 40138 Bologna, Italy; paolo.castellucci@aosp.bo.it (P.C.); stefano.fanti@aosp.bo.it (S.F.); 7UOC di Radioterapia Oncologica, Dipartimento Diagnostica per Immagini, Radioterapia Oncologica ed Ematologia, Fondazione Policlinico Universitario Agostino Gemelli IRCCS, 00168 Roma, Italy; luca.tagliaferri@policlinicogemelli.it; 8Division of Gynecologic Oncology, IRCCS Azienda Ospedaliero—Universitaria di Bologna, 40138 Bologna, Italy; 9Department of Medical Physics, IRCCS Azienda Ospedaliero—Universitaria di Bologna, 40138 Bologna, Italy; lidia.strigari2@unibo.it; 10Diagnostic and Interventional Radiology, IRCCS Istituto Ortopedico Rizzoli, 40136 Bologna, Italy; alberto.bazzocchi@ior.it; 11Service of Radiology, Imaging Institute of Southern Switzerland, Ente Ospedaliero Cantonale (EOC), CH-6500 Lugano, Switzerland; stefania.rizzo@eoc.ch; 12Medical Physics Unit, Responsible Research Hospital, 86100 Campobasso, Italy; savino.cilla@responsible.hospital

**Keywords:** locally advanced cervical cancer (LACC), prognostic models, hemoglobin, ECOG performance status, tumor size, inflammatory indices, total blood protein, sarcopenic obesity, machine learning, LASSO regression, CART model, local control (LC), metastasis-free survival (MFS), disease-free survival (DFS), overall survival (OS), receiver operating characteristic (ROC), area under the curve (AUC), predictive accuracy, immunotherapy, eosinophilia

## Abstract

**Background/Objective**: This study aimed to assess the prognostic significance of pretreatment nutritional and systemic inflammatory indices (IIs), and body composition parameters in patients with locally advanced cervical cancer (LACC) treated with chemoradiation and brachytherapy. The goal was to identify key predictors of clinical outcomes, such as local control (LC), metastasis-free survival (MFS), disease-free survival (DFS), and overall survival (OS), using machine learning techniques. **Materials and methods**: A retrospective analysis of 173 patients with LACC treated between 2007 and 2021 was conducted. The study utilized machine learning techniques, including LASSO regression and Classification and Regression Tree (CART) analysis, to identify significant predictors of outcomes. Clinical data, tumor-related parameters, and treatment factors, along with IIs and body composition metrics (e.g., sarcopenic obesity), were incorporated into the models. Model performance was evaluated using ROC curves and AUC values. **Results**: Among 173 patients, hemoglobin (Hb) levels, ECOG performance status, and total protein emerged as primary prognostic indicators across multiple endpoints. For 2-year LC, patients with Hb > 11.9 g/dL had a rate of 95.1% compared to 73.6% in those with lower levels, with further stratification by ECOG status, ANRI, and total protein refining predictions. For 5-year LC, rates were 83.1% for Hb > 11.5 g/dL and 43.3% for lower levels. For 2-year MFS, ECOG 0 patients had an 88.1% rate compared to 73.8% for ECOG ≥ 1. In 2-year OS, Hb > 11.9 g/dL predicted a 95.1% rate, while ≤11.9 g/dL correlated with 74.0%. IIs (ANRI, SIRI, MLR) demonstrated predictive value only within specific patient subgroups defined by the primary prognostic indicators. The model showed strong predictive accuracy, with AUCs ranging from 0.656 for 2-year MFS to 0.851 for 2-year OS. **Conclusions**: These findings underscore the value of integrating traditional prognostic factors with emerging markers to enhance risk stratification in LACC. The use of machine learning techniques like LASSO and CART demonstrated strong predictive capabilities, highlighting their potential to refine individualized treatment strategies. Prospective validation of these models is warranted to confirm their utility in clinical practice.

## 1. Introduction

Cervical cancer continues to be one of the most common cancers worldwide, especially in developing regions [1]. For locally advanced cervical cancer (LACC), concurrent chemoradiation (CRT) remains the standard treatment, with high rates of local control [2]. However, approximately one-third of patients still experience treatment failure after CRT, underscoring the need for better prognostic tools [3,4]. Several prognostic models for LACC have been developed, focusing on factors like tumor size, histology, lymph node involvement, and FIGO stage, all of which have shown significant correlations with overall survival (OS) [5,6]. Anemia is also a recognized negative prognostic factor in this population [7,8,9,10].

Recent research has explored systemic inflammatory indices (IIs) as potential prognostic markers. Elevated neutrophil-lymphocyte ratio (NLR) and platelet-lymphocyte ratio (PLR) have been associated with poorer disease-free survival (DFS) and OS, as well as reduced response to CRT in LACC [11,12,13,14,15,16,17,18,19,20,21,22,23,24,25,26,27]. However, most studies focus on single markers or limited variables, often without accounting for confounders. To address this gap, we analyzed a broad range of pretreatment nutritional and systemic IIs in LACC patients undergoing CRT, integrating clinical, nutritional, tumor-related, and treatment-related data.

Our previous analysis examined IIs like NLR, PLR, monocyte-lymphocyte ratio (MLR), and systemic immune inflammation index (SII). Significant correlations were found between poor outcomes and factors like older age, higher FIGO stage, lower hemoglobin (Hb) levels, larger tumor size, and higher body mass index (BMI). Notably, higher SII values were linked with lower distant metastasis-free survival (DMFS), though other inflammatory markers showed no significant association with DFS or OS in multivariate analysis [28].

In a follow-up study on the same cohort, we aimed to validate published cut-offs for systemic IIs, such as NLR and PLR. While univariate analysis identified several associations, multivariate analysis confirmed Hb levels, radiation dose, and age as the most important predictors of OS, with no significant impact from IIs [29]. Another analysis explored anthropometric factors, revealing sarcopenic obesity as a predictor of DFS and OS in both univariate and multivariate analysis [30].

Given the complexity of prognostic modeling in cervical cancer and the limitations of traditional statistical methods, the application of new variable selection methods and AI-based techniques, such as Least Absolute Shrinkage and Selection Operator (LASSO) regression and CART (Classification and Regression Trees), offers significant potential advantages. These advanced methods can handle large datasets with numerous variables, improve model accuracy by preventing overfitting, and allow for more individualized predictions. LASSO, by selecting only the most relevant predictors, enhances the robustness of the model, while CART provides an intuitive, decision-tree-based structure that is easy to interpret in clinical practice. This supervised machine-learning approach is particularly well-suited for incorporating a wide range of prognostic factors, from clinical and tumor characteristics to IIs and anthropometric parameters.

In light of these advancements, we conducted a comprehensive analysis using LASSO and CART methodologies to refine the identification of key prognostic factors in LACC. By using these AI-based tools, our study aims to improve upon existing predictive models, offering a more refined and clinically applicable prognostic approach in LACC.

## 2. Materials and Methods

### 2.1. Aim and Design of the Study

This study aimed to evaluate the prognostic significance of pretreatment systemic IIs, along with body composition parameters, in patients with LACC. Confounding factors, including patient characteristics, tumor features, functional imaging (FDG-PET), and treatment data, were also considered. The study focused on four clinical endpoints: local control (LC), DMFS, DFS, and OS. We retrospectively analyzed patients treated between July 2007 and July 2021, who were enrolled in an observational study approved by the local Ethical Committee (ESTHER study, code CE 973/2020/Oss/AOUBo). All patients provided informed consent, and none were excluded to reflect real-world clinical practice.

### 2.2. Staging, Treatment, and Follow-Up

Patients were retrospectively staged according to the 2018 International Federation of Gynecology and Obstetrics (FIGO) system. Treatment consisted of definitive CRT, combining external beam radiotherapy (EBRT) to the pelvis (45–50 Gy) with intracavitary brachytherapy to achieve a total equivalent dose of 85–90 Gy at the tumor site. Clinical and planning target volumes included the gross tumor, uterus, parametria, upper third of the vagina, and pelvic nodes, with a boost to nodal metastases when present. A daily patient set-up check was conducted, initially using an electronic portal imaging device and later cone-beam CT [31]. Chemotherapy involved weekly intravenous cisplatin (40 mg/m^2^). Follow-up included physical exams every three months for two years and then every six months for the next three years, with thoracic-abdominal-pelvic CT scans as clinically indicated or every six months in the first two years and annually thereafter.

### 2.3. Evaluated Parameters

#### 2.3.1. Clinical Data

Details of evaluated parameters (Appendix A) have been previously published [28,29,30] and are summarized below:

Patient-related data: Age, body mass index (BMI), standard blood tests, and complete blood count were recorded before CRT.

Tumor-related data: Tumor characteristics included histological type, FIGO stage (2018), tumor and nodal stages, and maximum tumor diameter.

Treatment-related data: EBRT technique and dose, brachytherapy dose, total tumor dose, and overall treatment time (EBRT plus brachytherapy) were documented.

#### 2.3.2. Inflammatory Indices

Various IIs were analyzed, including NLR, PLR, MLR, SII, and others (Appendix A), all calculated from routine blood tests prior to CRT.

#### 2.3.3. Body Composition Parameters

Body composition parameters, such as BMI, sarcopenia, and sarcopenic obesity, were assessed. BMI was categorized according to standard cut-offs [32], and pretreatment CT scans were used to determine skeletal muscle area at the level of the third lumbar vertebra [33]. Sarcopenia was defined using skeletal muscle index, while sarcopenic obesity was defined as the coexistence of sarcopenia and obesity (BMI ≥ 30 kg/m^2^), following established criteria [34,35] (Appendix A).

#### 2.3.4. Functional Imaging

Maximum standardized uptake value (SUV-max) from FDG-PET was recorded to assess FDG avidity, calculated as tracer uptake per unit of injected activity and patient weight.

### 2.4. Machine Learning Modeling and Statistical Analysis

LC was measured from CRT initiation to local recurrence, DMFS from CRT start to distant metastasis, DFS from treatment start to any treatment failure, and OS from CRT initiation to death or last follow-up. Variables with non-zero coefficients in the LASSO regression model were selected as key prognostic variables. These variables were then used to create CART models to predict clinical outcomes. CART is a data mining tool that uses decision trees to automatically identify patterns and relationships within data, even in large datasets. The CART model is represented as a binary tree, where each root node stands for an input feature and a split point on that feature. Predicted output variable is present in the tree’s leaf nodes. Model performance was assessed using Receiver Operating Characteristic (ROC) curves and the Area Under the Curve (AUC) values. The dataset was split into training and validation cohorts at a 70:30 ratio, with fivefold cross-validation applied to evaluate model accuracy.

## 3. Results

### 3.1. Patients’ Characteristics

A total of 173 patients were included in this analysis. The patient characteristics are detailed in Table 1. The median follow-up period was 36 months, ranging from 3 to 151 months. Outcomes were evaluated at 2 years and 5 years in 156 and 113 patients, respectively.

### 3.2. Predictive Model

#### 3.2.1. Local Control

In the subgroup of patients with Hb levels > 11.9 g/dL, the 2-year LC rate was 95.1%, while in patients with Hb levels ≤ 11.9 g/dL, the 2-year LC rate was 73.6%. Within this group, patients with an ECOG performance status of 0 had a 2-year LC rate of 85.7%, while those with ECOG ≥ 1 had a lower LC rate of 60.0%. Further stratification of the ECOG 0 group showed that patients with total protein levels > 7.1 g/dL achieved a 2-year LC rate of 100%, compared to 73.3% in those with total protein levels ≤ 7.1 g/dL. In the ECOG ≥ 1 group, patients with an ANRI ≤ 3.5 had a 2-year LC rate of 47.1%, whereas those with ANRI > 3.5 had a higher LC rate of 87.5% (Figure 1A). 

Regarding 5-year LC, in patients with Hb levels ≤ 11.5 g/dL, the 5-year LC rate was 43.3%, compared to 83.1% in those with Hb levels > 11.5 g/dL. Within the Hb ≤ 11.5 g/dL group, patients with ECOG 0 had a 5-year LC rate of 70.0%, whereas those with ECOG ≥ 1 had a significantly lower rate of 28.6%. Further stratification of the ECOG ≥ 1 group revealed that patients with AST levels ≤ 13 U/L had a 0.0% 5-year LC rate, while those with AST levels > 13 U/L had a higher rate of 46.2% (Figure 1B). 

#### 3.2.2. Metastasis-Free Survival

For the 2-year MFS, patients with ECOG 0 had a 2-year MFS rate of 88.1%, while those with ECOG ≥ 1 had a lower rate of 73.8%. In the ECOG ≥ 1 group, patients with FIGO stage IB-IIIB had a 2-year MFS rate of 86.7%, whereas those with FIGO stage IIIC1-IIIC2 or IVA had a lower rate of 62.9% (Figure 2A). 

For the 5-year MFS, patients with clinical nodal stage 2 had a 5-year MFS rate of 41.2%, while those with clinical nodal stage 0–1 had a higher rate of 71.6%. Within the N stage 0–1 group, patients with ECOG ≥ 1 had a 5-year MFS rate of 59.5%, compared to 81.8% in those with ECOG 0. In the ECOG 0 group, patients with total protein levels ≤ 6.9 g/dL had a 5-year MFS rate of 56.3%, whereas those with total protein levels > 6.9 g/dL had a markedly higher rate of 96.4% (Figure 2B). 

#### 3.2.3. Disease-Free Survival

For the 2-year DFS, patients with ECOG 0 had a DFS rate of 82.4%, while those with ECOG ≥ 1 had a lower rate of 60.6%. In the ECOG 0 group, further stratification based on red blood cell (RBC) count showed that patients with RBC ≤ 4.6 × 10^6^/µL had a DFS rate of 76.4%, whereas those with RBC > 4.6 × 10^6^/µL had a higher DFS of 93.3%. Among those with RBC ≤ 4.6 × 10^6^/µL, if the histological type was squamous cell carcinoma (SCC), the DFS was 82.2%, compared to 50.0% for other histological types. For patients with RBC > 4.6 × 10^6^/µL, those with eosinophil levels > 0.1 × 10^3^/µL had an excellent DFS rate of 100%, while those with eosinophils ≤ 0.1 × 10^3^/µL had a DFS of 75.0%. In the ECOG ≥ 1 group, further analysis revealed that patients with Hb ≤ 13 g/dL had a 2-year DFS of 48.9%, compared to 83.3% for those with Hb > 13 g/dL. Among patients with Hb ≤ 13 g/dL, those with FIGO stage IIIC1-IIIC2 or IVA had a DFS of 37.5%, while those with FIGO stage IB-IIIB achieved a DFS of 73.3%. For patients with Hb > 13 g/dL, stratification by SIRI showed that if SIRI ≤ 24.9, the DFS was 71.4%, but for patients with SIRI > 24.9, the DFS was 100% (Figure 3A). 

For the 5-year DFS, patients with ECOG ≥ 1 had a DFS rate of 41.1%, while those with ECOG 0 had a better outcome with a DFS of 66.7%. In the ECOG ≥ 1 group, those with FIGO stage IIIC1-IIIC2 or IVA had a DFS rate of 30.1%, compared to 78.6% for patients with FIGO stage IB-IIIB. In the ECOG 0 group, stratification by total protein levels showed that patients with total protein ≤ 6.9 g/dL had a DFS of 47.6%, while those with total protein > 6.9 g/dL had a DFS of 77.8%. Within this higher total protein group, those with nodal stage 1 had a DFS of 100%, compared to 65.2% for those with nodal stage 2 (Figure 3B).

#### 3.2.4. Overall Survival

For the 2-year OS, patients with Hb levels ≤ 11.9 g/dL had an OS rate of 74.0%, whereas those with Hb > 11.9 g/dL had a significantly higher OS rate of 95.1%. Among patients with Hb ≤ 11.9 g/dL, those with a maximum tumor diameter of ≤49 mm had an OS rate of 90.5%, while patients with a tumor diameter >49 mm had a lower OS rate of 62.1%. In the group with Hb > 11.9 g/dL, stratification by monocyte-lymphocyte ratio (MLR) showed that patients with MLR ≤ 0.2 had a 100% OS, while those with MLR > 0.2 had an OS of 91.8% (Figure 4A).

For the 5-year OS, patients with ECOG ≥ 1 had an OS rate of 52.1%, whereas those with ECOG 0 had a higher OS of 78.8%. Within the ECOG 0 group, patients with a maximum tumor diameter > 51 mm had an OS of 57.1%, while those with a tumor diameter ≤ 51 mm had an OS of 86.8%. Furthermore, among the patients with a tumor diameter ≤ 51 mm, those with total protein levels > 6.9 g/dL had a 100% OS, while those with total protein ≤ 6.9 g/dL had a lower OS of 66.7% (Figure 4B).

### 3.3. Receiver Operating Characteristic and Area Under the Curve

The predictive accuracy of the models, as assessed by the AUC of the ROC curves, varied across different endpoints and time frames. For LC, the model demonstrated strong predictive accuracy, with an AUC of 0.813 and 0.788 for 2-year LC and a slightly lower AUC of 0.739 and 0.727 for 5-year LC, in the training and validation cohorts, respectively (Figure 5a,b). For MFS, the model showed a moderate AUC of 0.709 and 0.656 for 2-year MFS and an improved AUC of 0.733 and 0.686 for 5-year MFS in the training and validation cohorts, respectively (Figure 5c,d). DFS predictions achieved an AUC of 0.796 (0.758 in the validation set) for 2-year DFS and 0.773 (0.746 in the validation set) for 5-year DFS, indicating robust predictive power for this endpoint (Figure 5e,f). Finally, the model for OS demonstrated strong accuracy with an AUC of 0.851 and 0.846 for 2-year OS but a lower AUC of 0.711 and 0.678 for 5-year OS in the training and validation cohorts, respectively, indicating slightly reduced predictive performance over the longer term (Figure 5g,h).

## 4. Discussion

For over two decades, the standard treatment for LACC has been concomitant cisplatin-based CRT followed by a brachytherapy boost. Despite various attempts to improve outcomes through treatment intensification, such as the addition of adjuvant chemotherapy, no significant improvements have been achieved [36]. However, recent data from the ENGOT-cx11/GOG-3047/KEYNOTE-A18 randomized trial demonstrated a marked improvement in progression-free survival among patients with “high-risk” LACC who received pembrolizumab alongside CRT, both concurrently and adjuvantly [37]. Furthermore, findings from this trial, along with results from the CALLA trial [38], highlighted that subgroup analyses, particularly those focusing on patients at the highest risk of progression or death based on disease stage, suggested that these individuals benefited most from the combination of CRT and immunotherapy. These findings underscore the critical need for predictive models that can effectively identify LACC patients at the highest risk of treatment failure, enabling more personalized and targeted therapeutic approaches.

This study aimed to assess the prognostic significance of pretreatment systemic inflammatory markers, along with body composition parameters, in patients with LACC treated with CRT and brachytherapy. A comprehensive retrospective analysis was conducted on 173 patients, evaluating clinical, tumor, and treatment-related factors. Prognostic models were developed using machine learning techniques, including LASSO regression and CART analysis, to identify key predictors of outcomes, such as LC, MFS, DFS, and OS. The results demonstrated that factors such as Hb levels, ECOG performance status, and tumor size played critical roles in predicting survival outcomes, with models showing moderate to strong predictive accuracy across various endpoints.

The ROC AUC analysis showed strong predictive accuracy for short-term outcomes, with the best performance in 2-year OS (AUC 0.851 in training and 0.846 in validation sets), indicating robust model discrimination for short-term survival. However, 5-year OS predictions were less accurate, with AUCs of 0.711 and 0.678, suggesting areas for improvement in long-term survival prediction. For local control, both 2-year (AUC 0.813 and 0.788) and 5-year (AUC 0.739 and 0.727) models performed reliably, albeit with a decline over time. DFS models maintained solid accuracy, with AUCs of 0.796 (training) and 0.758 (validation) at 2 years and slight drops at 5 years. MFS prediction accuracy was modest, particularly at 2 years (AUC 0.709 and 0.656), improving at 5 years (AUC 0.733 and 0.686). These findings support the model’s effectiveness in short-term prognostication while highlighting areas for improvement in long-term prediction.

Based on the comparison of recent studies (Table 2) [28,29,30,39,40,41,42,43,44], several considerations can be drawn. The present study, similar to others such as Hua L [40] and Xu C [41], utilized advanced modeling techniques (LASSO, CART) to develop predictive models, aligning with current trends in leveraging machine learning approaches to handle complex, multidimensional clinical data. Studies like Abdalvand N [42] also underscore the strength of machine learning models, particularly Random Forest, in enhancing prediction accuracy for treatment outcomes.

Across studies, traditional clinical factors such as Hb levels, tumor size, and ECOG status emerged as key predictors. This is consistent with findings from Medici et al. [29], where classical prognostic factors outperformed systemic IIs in predicting survival. While systemic indices, such as NLR and PLR, have been explored, their prognostic power remains inferior to traditional clinical factors [28].

In fact, while systemic inflammatory indices (IIs) were highlighted in studies like those by Luo et al. [39] and Leetanaporn et al. [43], our findings, alongside others [29], showed that IIs generally lack a significant standalone correlation with OS or DFS across the full patient cohort. However, when considering specific patient subgroups, these indices showed prognostic relevance, particularly for short-term outcomes and distinct patient characteristics. For instance, ANRI was associated with a 2-year LC rate in patients with Hb ≤ 11.9 g/dL and ECOG performance status ≥ 1, while SIRI correlated with a 2-year DFS in patients with ECOG ≥ 1 and Hb > 13.0 g/dL. Additionally, MLR was a predictor for 2-year OS in patients with Hb > 11.9 g/dL. These results suggest that the predictive power of IIs may depend on nuanced patient profiles and combinations of clinical factors. This finding highlights the complex role of IIs in prognostication for LACC, implying that their utility may lie in complementing traditional markers within specific clinical contexts, rather than as independent predictors. Further research is warranted to explore these interactions and validate the role of IIs in personalized treatment planning.

Interestingly, total blood protein levels emerged as a significant prognostic factor, influencing multiple short- and long-term outcomes, including 2-year LC, 5-year MFS, 5-year DFS, and 5-year OS. This suggests that total protein levels, reflecting both nutritional and inflammatory status, could serve as a surrogate marker in LACC. In particular, the consistent association across various clinical endpoints highlights its potential as an important marker, as previously reported [45].

Some partially unexpected results emerged from the analysis. Notably, the 5-year LC rate was 0% in patients with AST levels below 13, within the subgroup of patients with Hb levels ≤ 11.5 g/dL and ECOG performance status ≥ 1. This suggests that extremely low AST levels, typically associated with impaired liver function, may be indicative of worse LC outcomes in this already high-risk group. In general, it can be noted that several studies indicate that lower AST levels are associated with better survival outcomes [46], while others suggest that low levels in certain contexts could also reflect underlying issues that may complicate prognosis. For example, a study focusing on non-small cell lung cancer found that patients with preoperative AST levels of ≤19 U/L had shorter OS compared to those with higher levels (AST > 19 U/L) (*p* = 0.006) [47].

Conversely, the analysis revealed a strikingly favorable 2-year DFS rate of 100% in patients with eosinophil counts greater than 0.1, within the subgroup of non-anemic patients with ECOG 0. Other studies have suggested a positive correlation between eosinophilia and improved prognosis in non-small cell lung cancer [48], melanoma [49], and renal cell carcinoma [50]. These findings are thought to be related to the role of eosinophils in the tumor microenvironment and systemic immune response, both of which can influence cancer progression and treatment outcomes.

Unique to the present and one previous study [30] is the incorporation of body composition parameters, such as sarcopenic obesity, which was shown in Medici et al.’s study to be an independent predictor of worse outcomes. However, in our study, we did not observe the same correlation between sarcopenic obesity and prognosis. This discrepancy may be attributed to differences in the statistical methods applied, such as the use of machine learning techniques like LASSO and CART in our analysis, which might assess interactions between variables differently compared to traditional regression models. This highlights the importance of method selection in identifying prognostic factors and suggests that further investigation is needed to confirm the role of body composition in treatment planning.

In terms of predictive accuracy, the present study achieved moderate to strong AUCs for outcomes, with an AUC greater than 0.80 for both 2-year LC and OS. Other studies, such as Xu C. [41] and Zang L. [44], also achieved similar strong predictive accuracy (AUCs of 0.83–0.87), reinforcing the reliability of advanced models in improving prognostication in LACC.

In summary, while traditional prognostic factors continue to play a significant role in survival prediction, novel factors, such as body composition and systemic inflammation indices, require further exploration. Machine learning techniques, as seen in our study and others, provide a valuable tool for refining predictive models and enhancing personalized treatment in LACC.

However, several limitations should be acknowledged. First, this was a retrospective study, which may introduce inherent biases related to patient selection and data availability. Moreover, overfitting is a risk when employing advanced models like LASSO and CART. To address this, we implemented several strategies. First, we partitioned our dataset into separate training (70%) and validation (30%) cohorts. Second, we employed fivefold cross-validation in the training set to tune model hyperparameters, detect potential overfitting, and evaluate predictive stability. Finally, we reported the model’s performance metrics separately for both the training and validation sets to ensure transparent documentation of its behavior on unseen data. We acknowledge that techniques such as Elastic Net regression may provide a balanced approach by penalizing correlated features less severely than LASSO alone, thus potentially capturing multifactorial interactions better. Future efforts will explore these methods and further expand the range of predictive variables (e.g., genetic and molecular factors) to improve risk stratification in LACC.

Additionally, although we aimed to conduct a comprehensive analysis, certain established prognostic factors were unavailable in our dataset. For instance, the squamous cell carcinoma antigen (SCC), which is valuable both for follow-up monitoring [51] and prognostic prediction [52], was excluded from the analysis due to the limited number of patients with available data.

To the best of our knowledge, ours is the first experience that jointly analyzes clinical parameters, therapeutic parameters, inflammation indices, functional imaging results, and body composition parameters in cervical cancer. This unique, comprehensive approach has limited our ability to establish collaborations with other centers for external validation at this stage. We acknowledge that the single-institution nature of our dataset may limit the generalizability of our findings. In future prospective multicenter collaborations, we plan to incorporate broader demographic data and additional patient populations to strengthen the robustness and applicability of our model. Additionally, further research is needed to explore the integration of novel biomarkers, including more advanced systemic IIs and genetic or molecular data, to enhance prognostication. The role of body composition metrics, such as sarcopenic obesity, should also be investigated in larger cohorts to solidify their place in treatment planning. Finally, the application of machine learning techniques should continue to evolve, potentially incorporating dynamic, real-time data to further personalize treatment approaches for LACC patients.

## 5. Conclusions

In conclusion, this study successfully applied advanced machine learning techniques, specifically LASSO and CART, to refine prognostic modeling in LACC. By integrating traditional prognostic factors such as Hb levels, ECOG performance status, and tumor size with emerging biomarkers and IIs, our models achieved moderate to high predictive accuracy across multiple clinical endpoints. Key findings include the independent prognostic significance of total protein levels, which emerged as a potential marker of both short- and long-term outcomes, and unexpected results regarding eosinophil and AST levels, underscoring their complex roles in LC and OS.

These findings suggest a pathway toward more individualized treatment approaches, as these predictive markers could assist clinicians in stratifying patients by risk and tailoring therapy more precisely. For example, recognizing the prognostic implications of Hb and total protein levels could help guide monitoring and intervention strategies for patients at higher risk of recurrence or metastasis. The observed value of integrating machine learning in prognostic modeling, which can accommodate complex interactions among diverse markers, suggests that such approaches may outperform traditional methods in certain clinical scenarios.

Future studies should validate these models prospectively and expand biomarker exploration to include genetic or molecular data [53,54,55,56], which may further refine prognostic accuracy. This research marks an important step toward predictive, personalized care in LACC, offering potential to improve patient outcomes by better identifying high-risk individuals and optimizing therapeutic interventions.

## Figures and Tables

**Figure 1 jpm-15-00153-f001:**
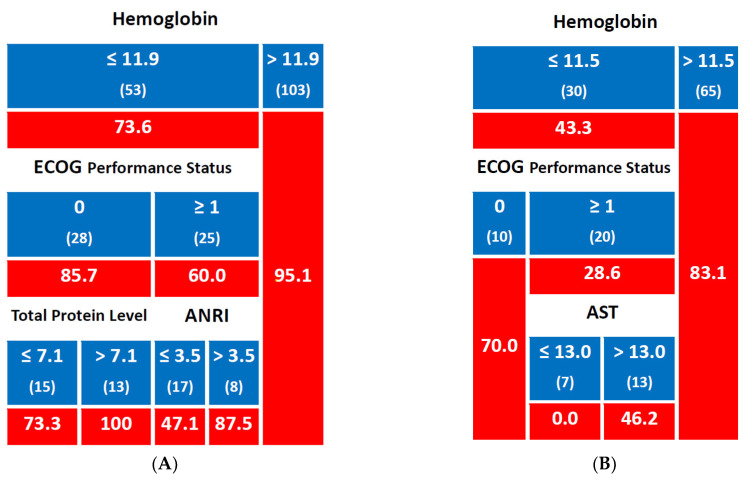
Predictive model for local control ((**A**): 2-year rate; (**B**): 5-year rate); number of patients in each group is shown in brackets; cells with a red background indicate percentage values; Hemoglobin and total protein levels are expressed in g/dL, while Aspartate Aminotransferase (AST) levels are expressed in U/L; ANRI stands for Albumin to Neutrophil Ratio Index.

**Figure 2 jpm-15-00153-f002:**
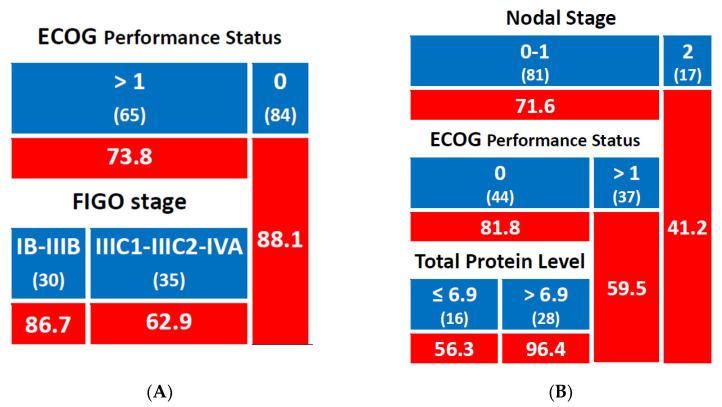
Predictive model for metastasis-free survival ((**A**): 2-year rate; (**B**): 5-year rate); number of patients in each group is shown in brackets; cells with a red background indicate percentage values. Total protein levels are expressed in g/dL.

**Figure 3 jpm-15-00153-f003:**
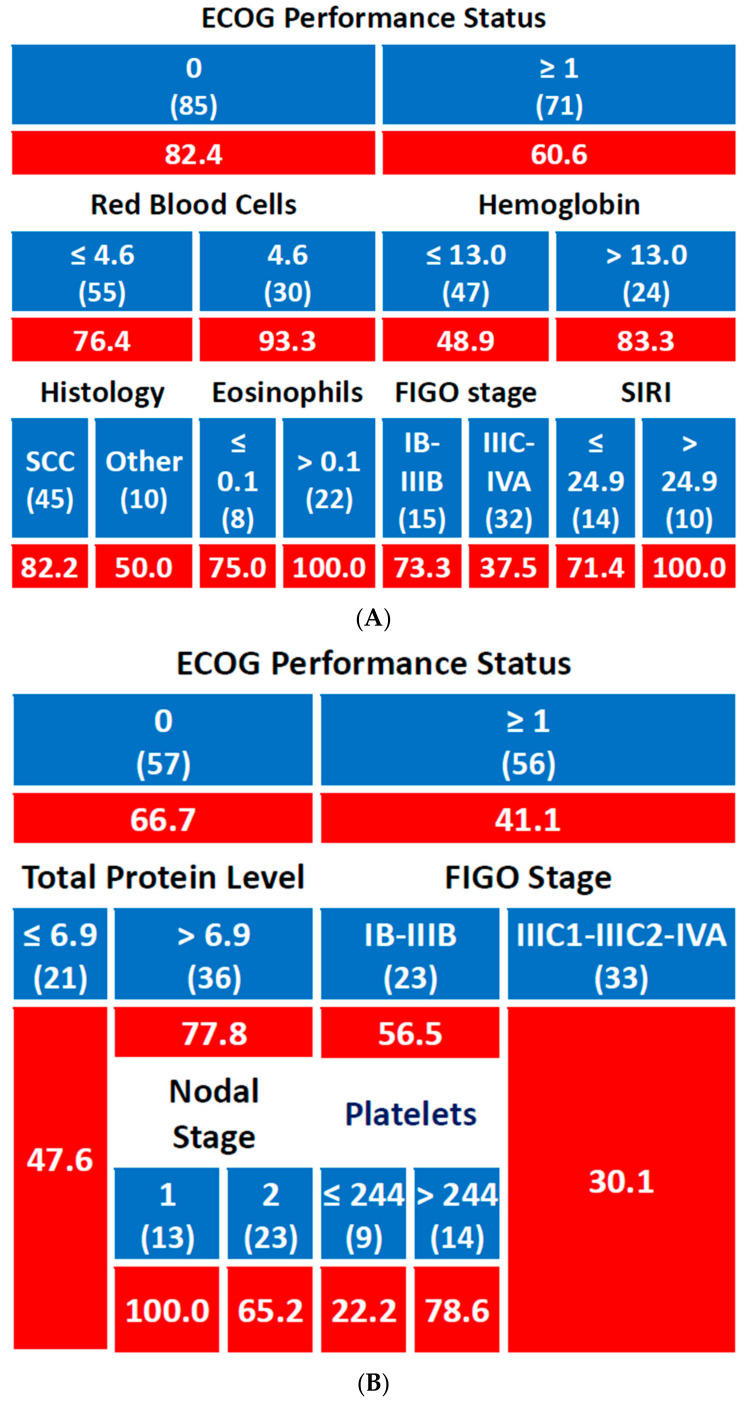
Predictive model for disease-free survival ((**A**): 2-year rate; (**B**): 5-year rate); number of patients in each group is shown in brackets; cells with a red background indicate percentage values. Red blood cell levels are expressed in 10^6^ cells/µL; hemoglobin levels in g/dL; eosinophil and platelets levels in 10^3^ cells/µL. SIRI stands for Systemic Inflammatory Response Index, and SCC stands for squamous cell carcinoma.

**Figure 4 jpm-15-00153-f004:**
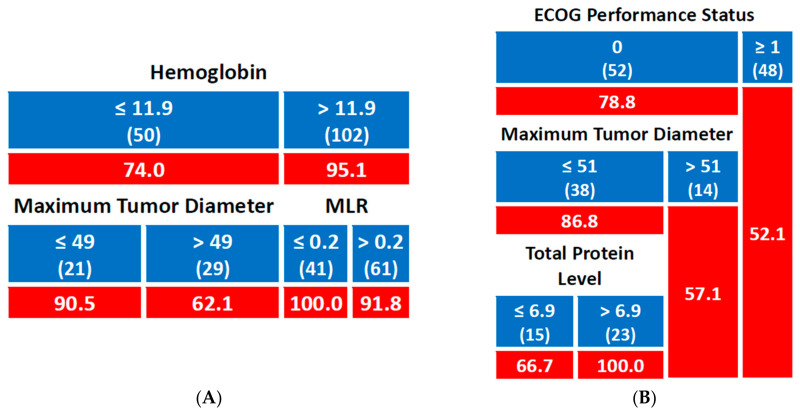
Predictive model for overall survival ((**A**): 2-year rate; (**B**): 5-year rate); number of patients in each group is shown in brackets; cells with a red background indicate percentage values. Hemoglobin and total protein levels are expressed in g/dL, while tumor diameter is expressed in mm.

**Figure 5 jpm-15-00153-f005:**
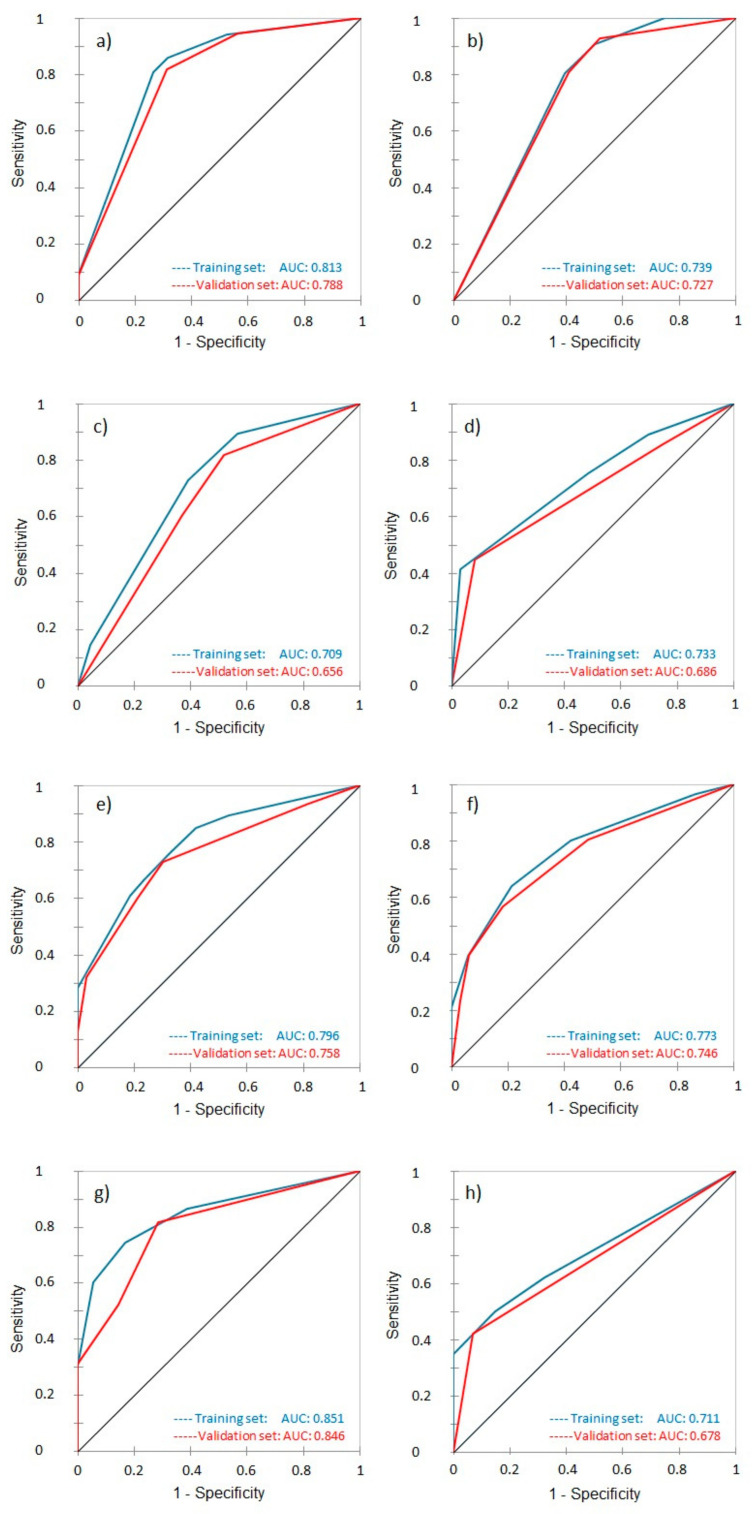
Receiver Operating Characteristic curves for the training (blue) and validation (red) datasets of the various predictive models: (**a**) 2-year and (**b**) 5-year local control, (**c**) 2-year and (**d**) 5-year metastasis-free survival, (**e**) 2-year and (**f**) 5-year disease-free survival, (**g**) 2-year and (**h**) 5-year overall survival.

**Table 1 jpm-15-00153-t001:** Patient characteristics.

Characteristic	Number of Patients (%)
Total patients	173 (100)
Median age (range), years	56 (27–85)
**Histological type**	
Squamous cell carcinoma	147 (85.0)
Adenocarcinoma	26 (15.0)
**FIGO stage**	
IB	1 (0.6)
IIA	3 (1.7)
IIB	73 (42.2)
IIIA	9 (5.2)
IIIB	3 (1.7)
IIIC1	39 (22.5)
IIIC2	22 (12.7)
IVA	23 (13.3)
**Radiotherapy technique**	
3D conformal radiotherapy	87 (50.3)
Intensity-modulated radiotherapy	66 (38.1)
Volumetric modulated arc therapy	20 (11.6)
**Median radiotherapy dose (range), Gy**	
Pelvic nodes (prophylactic)	46.0 (26.0–50.4)
Metastatic nodes	57.5 (52.5–61.0)
Brachytherapy boost	28.0 (4.0–42.0)

**Table 2 jpm-15-00153-t002:** Results of our analysis and selected recent studies.

Authors, Year	Aims	Methods	Results	Conclusions
Zang L, 2021 [41]	Develop a nomogram to predict OS in FIGO II-III CC treated with RT.	Retrospective study (469 patients). Cox regression and nomogram model creation.	C-index for nomogram = 0.71, better than FIGO staging.	Nomogram outperformed FIGO staging in predicting OS, offering a valuable clinical tool.
Leetanaporn K, 2022 [43]	Evaluate the predictive value of the HALP index on oncological outcomes in LACC patients.	Retrospective study (1588 patients). HALP cutoff identified using X-tile for survival model building.	HALP > 22.2 associated with better PFS and OS; improved model accuracy.	HALP is an independent predictor of survival, enhancing oncological outcome predictions.
Abdalvand N, 2022 [42]	Predict BRT response in LACC using clinical, physical, and dosimetric parameters via ML models.	Retrospective study (111 patients). ML models (LASSO, Ridge, SVM, Random Forest).	Random Forest models (AUC 0.82) outperformed reference models for response prediction.	ML models, especially Random Forest, improve BRT outcome prediction.
Ferioli M, 2023 [28]	Compare classical prognostic factors versus II in LACC.	Comprehensive retrospective analysis of IIs and survival (multivariate Cox).	Classical factors (age, tumor stage, Hb) were better predictors of OS than IIs.	Classical factors outperformed IIs for survival prediction in LACC patients.
Medici F, 2023 [29]	Assess the impact of systemic IIs on survival outcomes in LACC.	Retrospective study (173 patients). Multivariate Cox regression analysis of pretreatment IIs.	Hb levels, CRT dose, and age were significant predictors of OS; no IIs correlated with DFS or OS.	Classical prognostic factors outperform systemic IIs in predicting survival.
Luo Y, 2023 [39]	Develop a nomogram using TK1, inflammatory markers, and tumor markers to predict recurrence post-RT in intermediate-advanced CC.	Retrospective study (114 patients). Logistic regression for nomogram creation and validation (C-index and calibration curves).	TK1 and SCC antigen were independent predictors of recurrence (C-index 0.79).	Nomogram based on TK1 and inflammatory markers is more reliable than TNM staging for recurrence prediction.
Xu C, 2023 [41]	Develop a hybrid radiomics model to predict OS in CC patients receiving CCRT.	Retrospective study (367 patients). Handcrafted and DL-based radiomics features from CT for hybrid nomogram.	AUCs for OS = 0.83, 0.77, and 0.87 (1, 3, 5-year).	Hybrid radiomics model predicts OS effectively, aiding risk stratification in CC patients.
Hua L, 2024 [40]	Construct a survival prediction model for LACC patients treated with CCRT ± adjuvant chemotherapy.	Retrospective analysis (482 patients). Cox and LASSO regression for model building.	Validated risk factors for PFS and OS (AUC for OS = 0.94 at 1 year).	Supports accurate survival prediction and potential benefits of adjuvant chemotherapy for high-risk LACC.
Medici F, 2024 [30]	Evaluate sarcopenic obesity as a prognostic factor in CC outcomes.	Retrospective study (173 patients). Kaplan-Meier and Cox regression analysis.	Sarcopenic obesity was an independent predictor of worse DFS and OS.	Sarcopenic obesity is a strong prognostic factor and should be considered in treatment planning.
*Present study*	Evaluate prognostic significance of pretreatment, nutritional, systemic inflammatory markers, and body composition in LACC.	Retrospective analysis of 173 patients using LASSO and CART models to predict LC, MFS, DFS, and OS	Hemoglobin levels, ECOG status, and tumor size were key predictors of outcomes. ROC AUCs ranged from moderate to strong (AUC up to 0.851 for 2-year OS).	Predictive models effectively identified patients at a higher risk of poor outcomes, supporting personalized treatment strategies in LACC.

## Data Availability

The original data presented in the study are openly available in Zenodo Repository at https://doi.org/10.5281/zenodo.14944742.

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
