# Peer review of "Integrating Novel and Classical Prognostic Factors in Locally Advanced Cervical Cancer: A Machine Learning-Based Predictive Model (ESTHER Study)"

_jpm, 2025, doi:10.3390/jpm15040153_

Round 1
Reviewer 1 Report
Comments and Suggestions for Authors
The accuracy of the machine learning model described in this manuscript heavily depends on the quality and representativeness of the dataset. Authors have described it as specific demographics aspects that can be biased as far as global importance is concerned. It may contain missing/inconsistent values, the model's predictions may be unreliable. Please add some more data of diverse background and then go for the modeling.
Another point I have observed is about the machine learning models. As per my own experience, working with infections modeling, models are complex ones and are prone to overfitting, where the model performs well on training data but fails to generalize to new, unseen data. This could limit its clinical applicability of this model.
Also, the validation of this model needs rigorous applicability. If the model is not validated on independent, diverse datasets, its performance may not be reproducible in different clinical settings or populations, reducing its generalizability.
The idea and work are good, no doubt about it.
Comments on the Quality of English LanguageThe accuracy of the machine learning model described in this manuscript heavily depends on the quality and representativeness of the dataset. Authors have described it as specific demographics aspects that can be biased as far as global importance is concerned. It may contain missing/inconsistent values, the model's predictions may be unreliable. Please add some more data of diverse background and then go for the modeling.
Another point I have observed is about the machine learning models. As per my own experience, working with infections modeling, models are complex ones and are prone to overfitting, where the model performs well on training data but fails to generalize to new, unseen data. This could limit its clinical applicability of this model.
Also, the validation of this model needs rigorous applicability. If the model is not validated on independent, diverse datasets, its performance may not be reproducible in different clinical settings or populations, reducing its generalizability.
The idea and work are good, no doubt about it.
Author Response
We sincerely appreciate the thoughtful and constructive feedback provided by both reviewers. Your comments have been invaluable in helping us refine and strengthen our manuscript.
REVIEWER 1
Comment 1
The accuracy of the machine learning model described in this manuscript heavily depends on the quality and representativeness of the dataset. Authors have described it as specific demographics aspects that can be biased as far as global importance is concerned. It may contain missing/inconsistent values, the model predictions may be unreliable. Please add some more data of diverse background and then go for the modeling.
Response 1
We thank you for this valuable suggestion. Indeed, we recognize that the representativeness and completeness of any dataset can heavily influence the performance of machine learning models. As outlined in our Methods section, we have included a very large cohort that required years of work to collect, integrate, and calculate both the body composition and inflammation indices. To the best of our knowledge, ours is the first experience that jointly analyzes clinical parameters, therapeutic parameters, inflammation indices, functional imaging results, and body composition parameters in cervical cancer. This unique, comprehensive approach has limited our ability to establish collaborations with other centers for external validation at this stage. While we share your view that a more diverse patient population would be beneficial for broader generalizability, at this time it is not feasible for us to add further data, given the retrospective nature of our study.
Based on your comment, we have added the following text to the discussion section:
“To the best of our knowledge, ours is the first experience that jointly analyzes clinical parameters, therapeutic parameters, inflammation indices, functional imaging results, and body composition parameters in cervical cancer. This unique, comprehensive approach has limited our ability to establish collaborations with other centers for external validation at this stage. We acknowledge that the single-institution nature of our dataset may limit the generalizability of our findings. In future prospective multicenter collaborations, we plan to incorporate broader demographic data and additional patient populations to strengthen the robustness and applicability of our model.”
We thank you again for your insightful comment and appreciate your understanding regarding the constraints of our current dataset. We are committed to expanding and diversifying our cohorts in future prospective studies.
Comment 2
Another point I have observed is about the machine learning models. As per my own experience, working with infections modeling, models are complex ones and are prone to overfitting, where the model performs well on training data but fails to generalize to new, unseen data. This could limit its clinical applicability of this model.
Response 2
Thank you for raising this important concern. Overfitting is indeed a risk when employing advanced models like LASSO and CART. To address this, we implemented several strategies. First, we partitioned our dataset into separate training (70%) and validation (30%) cohorts. Second, we employed fivefold cross-validation in the training set to tune model hyperparameters, detect potential overfitting, and evaluate predictive stability. Finally, we reported the model’s performance metrics separately for both the training and validation sets to ensure transparent documentation of its behavior on unseen data. We agree that external validation in independent cohorts would further enhance the clinical reliability of our models, and we will pursue such validation in future studies.
Based on your comment we have added the following text to the discussion section:
“Moreover, overfitting is a risk when employing advanced models like LASSO and CART. To address this, we implemented several strategies. First, we partitioned our dataset into separate training (70%) and validation (30%) cohorts. Second, we employed fivefold cross-validation in the training set to tune model hyperparameters, detect potential overfitting, and evaluate predictive stability. Finally, we reported the model’s performance metrics separately for both the training and validation sets to ensure transparent documentation of its behavior on unseen data.”
Thank you again for this valuable observation. We appreciate your emphasis on model robustness and generalizability, and we hope our added clarifications address your concern.
Comment 3
Also, the validation of this model needs rigorous applicability. If the model is not validated on independent, diverse datasets, its performance may not be reproducible in different clinical settings or populations, reducing its generalizability.
Response 3
We fully agree with your comment on the importance of rigorous external validation. As discussed in our responses above, this remains a key goal for our future work. Thank you again for underscoring this critical aspect.
Comment 4
The idea and work are good, no doubt about it.
Response 4
Thank you very much for your encouraging words. We truly appreciate your positive assessment of our study overall concept and methodology, it is motivating to receive such feedback.
Reviewer 2 Report
Comments and Suggestions for Authors
I appreciate the idea of study and the structure of manuscript but as far as these machine learning techniques like LASSO and CART are related to demonstrating their predictive capabilities, I am little bit concerned as most of the time LASSO tends to select only one variable among all other highly corelated ones that could be the chance of missing other important predictive markers. Moreover, in fields like cancer we have very complex interactions that involve genetic, environmental and clinical factors so in this case using linear models like LASSO may fail to capture effectively. I would suggest the author either possible mitigations like using Elastic net regression to balance their study or if it’s not possible to this time then they should at least discuss this possible limitations like these in their discussion section.
Author Response
We sincerely appreciate the thoughtful and constructive feedback provided by both reviewers. Your comments have been invaluable in helping us refine and strengthen our manuscript
REVIEWER 2
Comment 1
I appreciate the idea of the study and the structure of manuscript, but as far as these machine learning techniques like LASSO and CART are related to demonstrating their predictive capabilities, I am a little bit concerned. Most of the time LASSO tends to select only one variable among all other highly correlated ones. This could lead to the chance of missing other important predictive markers. Moreover, in fields like cancer we have very complex interactions that involve genetic, environmental, and clinical factors, so using linear models like LASSO may fail to capture such complexity. I would suggest either possible mitigations (like using Elastic Net regression) or at least discuss limitations like these in the Discussion section.
Response 1
Thank you for highlighting these valid points. We agree that LASSO tendency to force sparsity can sometimes exclude correlated predictors that might contain complementary information. We initially selected LASSO for its strong feature-selection properties and its ability to minimize overfitting in high-dimensional datasets. However, as you correctly note, cancer is characterized by intricate biological, molecular, and environmental interactions, which often go beyond purely linear relationships.
At this time, we are not able to re-run the entire modeling using Elastic Net or other advanced frameworks, due to time constraints and the extensive analyses already performed. We understand and acknowledge the potential benefits of such an approach, and we plan to explore alternative models like Elastic Net in our future work. We appreciate your thoughtful suggestion and your understanding of the limitations we currently face. Nonetheless, we have revised our Discussion to acknowledge and clarify these limitations. Specifically, we have added:
“We acknowledge that techniques such as Elastic Net regression may provide a balanced approach by penalizing correlated features less severely than LASSO alone, thus potentially capturing multifactorial interactions better. Future efforts will explore these methods and further expand the range of predictive variables (e.g., genetic and molecular factors) to improve risk stratification in locally advanced cervical cancer.”
Thank you once again for your constructive feedback: it has helped us improve the clarity and depth of our discussion.
Round 2
Reviewer 1 Report
Comments and Suggestions for Authors
Accepted
Comments on the Quality of English LanguageAccepted